# Antioxidant Efficacy of Green-Synthesized Silver Nanoparticles Promotes Wound Healing in Mice

**DOI:** 10.3390/pharmaceutics15051517

**Published:** 2023-05-17

**Authors:** Vajravathi Lakkim, Madhava C. Reddy, VijayaDurga V. V. Lekkala, Veeranjaneya Reddy Lebaka, Mallikarjuna Korivi, Dakshayani Lomada

**Affiliations:** 1Department of Genetics, Yogi Vemana University, Kadapa 516005, AP, India; vajravathi@gmail.com (V.L.); lvijayadurga123@gmail.com (V.V.V.L.); 2Department of Biotechnology and Bioinformatics, Yogi Vemana University, Kadapa 516005, AP, India; cmreddy@yogivemanauniversity.ac.in; 3Department of Microbiology, Yogi Vemana University, Kadapa 516005, AP, India; lvereddy@yvu.edu.in; 4Exercise and Metabolism Research Center, College of Physical Education and Health Sciences, Zhejiang Normal University, Jinhua 321004, China

**Keywords:** wound healing, antioxidants, collagen, *Azadirachta indica*, *Catharanthus roseus*, silver nanoparticles

## Abstract

Developing an efficient and cost-effective wound-healing substance to treat wounds and regenerate skin is desperately needed in the current world. Antioxidant substances are gaining interest in wound healing, and green-synthesized silver nanoparticles have drawn considerable attention in biomedical applications due to their efficient, cost-effective, and non-toxic nature. The present study evaluated in vivo wound healing and antioxidant activities of silver nanoparticles from *Azadirachta indica* (AAgNPs) and *Catharanthus roseus* (CAgNPs) leaf extracts in BALB/c mice. We found rapid wound healing, higher collagen deposition, and increased DNA and protein content in AAgNPs- and CAgNPs (1% *w*/*w*)-treated wounds than in control and vehicle control wounds. Skin antioxidant enzyme activities (SOD, catalase, GPx, GR) were significantly (*p* < 0.05) increased after 11 days CAgNPs and AAgNPs treatment. Furthermore, the topical application of CAgNPs and AAgNPs tends to suppress lipid peroxidation in wounded skin samples. Histopathological images evidenced decreased scar width, epithelium restoration, fine collagen deposition, and fewer inflammatory cells in CAgNPs and AAgNPs applied wounds. In vitro, the free radical scavenging activity of CAgNPs and AAgNPs was demonstrated by DPPH and ABTS radical scavenging assays. Our findings suggest that silver nanoparticles prepared from *C. roseus* and *A. indica* leaf extracts increased antioxidant status and improved the wound-healing process in mice. Therefore, these silver nanoparticles could be potential natural antioxidants to treat wounds.

## 1. Introduction

Wound healing, diabetic foot ulcers (DFUs), or the mechanism of burn injury are complex processes that require a well-coordinated communication among cytokines, growth factors, cell–matrix, and cell–cell interactions. Recently, nanotechnology has been widely used to treat various diseases, such as wounds, cardiovascular diseases, diabetes, and cancer [1,2]. The rise of acute and chronic wound-associated complications has led to the practice of alternate methods using nanotechnology, as is the case in the current scenario [3]. The physiological process of wound healing involves blood clot formation, inflammation, re-epithelialization, neovascularization, and tissue tightening [4,5]. Some chronic wounds may not completely heal through the normal stages of the healing process, and 40–50% of unhealed wounds need several weeks of proper treatment [6]. Reactive oxygen species (ROS) or free radicals, including superoxide anion, hydrogen peroxide (H_2_O_2_), and hydroxyl radical (·OH), are the key contributors to oxidative stress or cell damage [7,8]. Although an overwhelmed production of ROS leads to adverse effects, physiological concentrations of ROS play a significant role in the normal wound-healing process [9]. It has been stated that lower concentrations of ROS protect tissues against infections and stimulate wound healing by triggering required signaling cascades [10,11]. The accumulation of ROS primarily causes oxidative stress and pro-inflammatory status, leading to cell damage [12]. Subsequently, uncontrolled ROS production dysregulates the key steps in the wound-healing process [13]. The concentrations of H_2_O_2_, ranging from 100 to 250 μM, are considerable for the normal wound-healing process [11,14]. A study has shown that 10 μM concentration of H_2_O_2_ acts as a chemo-attractant required for the proliferation of endothelial cells and fibroblasts and 100 μM H_2_O_2_ stimulates angiogenesis through vascular endothelial grown factor (VEGF), while 500 μM H_2_O_2_ produces macrophage inflammatory protein-1α, leading to a pro-inflammatory status [15]. However, the antioxidants, which donate their electrons to unstable ROS, prevent taking electrons from the biomolecules, such as DNA, lipids, or proteins. Superoxide dismutase (SOD), catalase (CAT), glutathione peroxidase (GPx), and glutathione reductase (GR) are the main antioxidant enzymes that contribute to wound healing. Flavonoids, glutathione, vitamin C, and vitamin E are the non-enzymatic antioxidants [16,17] that also take place in the healing process. Either enzymatic or non-enzymatic antioxidants catalyze a series of complex reactions and thereby stabilize the ROS/free radicals. These antioxidant substances can maintain lower concentrations of ROS at wound areas, and promote the healing process [16]. Therefore, interest has grown regarding the usage of antioxidant substances for the treatment of wounds [18]. Taken together, it reveals that adequate antioxidant status is crucial in developing novel therapies to enhance the wound-healing process.

In this context, nanoparticles conjugated with natural antioxidants could be a possible effective intervention to promote wound-healing mechanisms. We have studied titanium dioxide nanotubes (TNT), and our data suggest that these materials have immunomodulation properties [19]. Furthermore, titanium dioxide nanomaterials exhibit anti-cancer properties and ameliorate autoimmune diseases in animal models [19,20,21,22]. Silver nanoparticles from plants are eco-friendly, non-toxic, and less expensive than nanoparticles synthesized from physical and chemical methods [23]. Among the noble metal nanoparticles, silver nanoparticles (AgNPs) have been increasingly used recently due to their exceptional biological properties [24]. Green-synthesized nanoparticles have capping molecules, which offer stabilization needed for biomedical applications [25,26,27]. Phytochemicals from *Azadirachta indica* (*A. indica*) leaf extracts are rich in naturally inherent oxygen quenchers, redox agents, and hydrogen donors that inactivate free radicals and/or trigger antioxidant enzymes to disturb the oxidation chain reaction [28]. The importance of silver nanoparticles in wound healing has been emphasized by Paladini et al. in 2019 [29]. Fitzmaurice et al. reviewed the functional role of antioxidants in wound healing [18]. AgNPs improve the wound-healing process by: (1) enhancing anti-inflammatory property (2) the antibacterial activity to a large number of bacterial strains, (3) incorporating AgNPs in wound dressings, and (4) surface modifications that aid in drug delivery [30,31]. The importance of antioxidant therapy in DFU has been reviewed by Zhang et al. in 2021 [32]. Green-synthesized AgNPs showed characteristic wound-healing properties by increasing keratinocyte proliferation and migration to damaged wound areas [33]. In our earlier study, we synthesized and characterized AgNPs from the leaf extracts of *Catharanthus roseus* (*C. roseus* CAgNPs) and *A. indica* (AAgNPs) and demonstrated their antibacterial and wound-healing properties [34]. However, the antioxidant role of CAgNPs and AAgNPs in wound healing remains unknown. Therefore, this study aimed to investigate the antioxidant efficacy of green-synthesized AgNPs in the wound-healing process using the excision wound model.

## 2. Materials and Methods

### 2.1. Chemicals and Materials

Phosphate-buffered saline, hydroxyproline, Ehrlich’s reagent, and proteinase K were purchased from the HiMedia Laboratories (Mumbai, India). Silver nitrate, quercetin, and DPPH were purchased from Sigma-Aldrich (St. Louis, MI, USA).

### 2.2. Preparation of Nanoparticles

Leaf extracts from *C. roseus* and *A. indica* were prepared and used to synthesize silver nanoparticles. Young leaves of *C. roseus* and *A. indica* were obtained from the Yogi Vemana University (14.473786° N, 78.711482° E) premises in Kadapa, Andhra Pradesh, India. The leaves were then cleaned thoroughly to remove the debris and other organic constituents, and later dried at 37 °C. Dried leaf (10 g) powder obtained from *C. roseus* and *A. indica* was suspended in water (100 mL) and boiled at 80 °C for an hour. The filtered leaf extract was mixed with silver nitrate to prepare the silver nanoparticles, as described earlier in the study of Lakkim et al. 2020 [34]. The dark brown color indicated the formation of silver nanoparticles.

### 2.3. Characterization of Nanoparticles

The silver nanoparticles synthesized from *C. roseus* and *A. indica* were characterized using X-ray diffraction (XRD), Fourier-transform infrared spectroscopy (FTIR), dynamic light scattering (DLS), scanning electron microscopy (SEM) with energy dispersive X-ray analysis (SEM-EDX), and transmission electron microscopy (TEM) analyses. The detailed protocols for each technique were fully explained in our previous publication [34].

### 2.4. In Vitro Free Radical Scavenging Assays

The free radical scavenging activity of AgNPs from *C. roseus* and *A. indica* was tested by DPPH (2, 2-diphenyl-1-picrylhydrazyl) assay. Quercetin, a known antioxidant, was used as a reference and blank using only phosphate buffer solution pH-7.4. Each experiment was performed in triplicate. The inhibition percentage was measured against blank. The percentage of DPPH radical scavenging activity was calculated according to the formula [35,36].
$$\mathrm{DPPH}\;\mathrm{radical}\;\mathrm{scavenging}\;\mathrm{activity}\;\left(\%\right)=\frac{\mathrm{Absorbance}\;\mathrm{of}\;\mathrm{control}-\mathrm{absorbance}\;\mathrm{of}\;\mathrm{sample}}{\mathrm{Absorbance}\;\mathrm{of}\;\mathrm{control}}\times 100$$

Next, the ABTS (2, 2′-azino-bis (3-ethylbenzothiazoline-6 sulfonic acid) scavenging activity of CAgNPs and AAgNPs was performed according to the previously published methods [37,38]. Quercetin was used as a standard, and distilled water was used as a blank. The percentage of ABTS radical scavenging activity with all treatments was calculated using the formula.
$$\mathrm{ABTS}\;\mathrm{radical}\;\mathrm{scavenging}\;\mathrm{activity}\;\left(\%\right)=[1-\frac{\left(\mathrm{Absorbance}\;\mathrm{of}\;\mathrm{sample}\right)}{\left(\mathrm{Absorbance}\;\mathrm{of}\;\mathrm{control}\right)}]\times 100$$

### 2.5. Grouping of Mice and Treatment

BALB/c mice were purchased from Mahaveera Enterprises (Hyderabad, India), and the wound experiment protocol was approved by the Institutional Animal Ethics Committee (IAEC) with CPCSEA registration no: 1841/GO/Re/S/51/CPCSEA. The mice were randomly assigned into five groups, as described in Lakkim et al., 2020 [34], which include Group I—control, Group II—positive control, Group III—vehicle control, Group IV—CAgNPs, and Group V—AAgNPs. The wound excision procedure was conducted according to the guidelines for the care and use of laboratory animals. The anesthesia dose was prepared for 5 mL, containing 2 mL ketamine (50 mg/mL), 0.5 mL xylazine (2%), and 2.5 mL saline (9%). The hair on the dorsal skin of the mice was removed with an artificial hair removal cream and disinfected with 70% ethanol. Mice were anesthetized with 40–50 µL of ketamine and xylazine mixture depending on the body weight, and a total thickness open excision wound was made with a 5 mm biopsy punch. The control group mice were treated with PBS, the positive control group was treated with betadine, and the vehicle control group was treated with vaseline. Betadine (povidone-iodine) ointment was used as a positive control because it is a widely accepted antiseptic and wound-healing agent [39]. Mice in CAgNPs and AAgNPs groups were treated with *C. roseus* AgNPs (1% *w*/*w*) and *A. indica* AgNPs (1% *w*/*w*), respectively. All respective substances in Group I, Group II, Group III, Group IV, and Group V were topically applied to the open excision wound. Following recovery from anesthesia, mice were housed in sanitized cages.

We pre-formulated 1 mg of green-synthesized AgNPs (CAgNPs or AAgNPs) ground in 1 gm of vaseline (1 mg AgNPs per 1 gm vaseline) and prepared a fine paste using a motor and pestle. This paste was used for topical application to wound surfaces on alternative days for a period of 10 days. Wound constriction was observed by monitoring the wound size on days 0, 2, 4, 6, 8, and 10. The wound closure area was measured alternatively using a digital vernier caliper. Wound-healing efficacy following treatment was represented as the wound contraction rate (WCR) percentage and calculated as described below. The percentage of wound healing in CAgNPs or AAgNPs groups was compared with controls.

The percentage of wound contraction rate = original wound area—actual wound area/original wound area × 100.

### 2.6. Biochemical Assays of Wound Skin Tissue

Biochemical assays were performed to estimate the collagen, DNA, and protein levels in the skin tissues (day 11) of BALB/c mice treated with green-synthesized silver nanoparticles. Hydroxyproline assay is widely used to estimate collagen. The hydroxyproline amount is expressed as mg hydroxyproline/g of dry skin tissue weight. The diphenylamine method for DNA estimation and Lowry’s method for protein estimation was used in wound-healing tissues of all treatment groups. Total protein was estimated by the Lowry method [40].

### 2.7. Histopathological Studies

Masson’s trichrome and hematoxylin and eosin (H&E) staining experiments were performed to evaluate the skin morphology and collagen deposition during wound healing in BALB/c mice treated with green-synthesized silver nanoparticles (CAgNPs and AAgNPs). Slides containing tissue sections were deparaffinized in xylene for 3 min, then immersed in graded percent of ethanol 100, 95, 80, and 70% for five minutes in each, followed by a thorough rinsing with distilled water. Further, slides were kept in hematoxylin for 1 min and washed excess stain with water. Slides were immersed in eosin for 30 s and then in graded percent of ethanol (100, 95, 80, and 70%). Sections were mounted with dibutylphalate polystyrene xylene (DPX) and examined under an inverted microscope (Make: Lawrence and Mayo, Model: CT3).

### 2.8. Estimation of Antioxidant Enzyme Status and Lipid Peroxidation in Wound Skin

Changes in antioxidant enzymes, including SOD, CAT, GPx, GR, glutathione-S- transferase (GST), and lipid peroxidation markers, were determined in the wound tissue of AgNPs-treated groups, and results were compared with control. The catalase activity in wound tissue was measured using the Aebi with minor modifications [41]. Catalase activity was determined using a 43.6 M cm^−^^1^ molar extinction coefficient. The moles of H_2_O_2_ degraded/mg protein/min are considered one unit of enzyme activity. GPx activity was determined using the method of Rotruck et al. with minor modifications [42]. GPx activity was measured in micromoles of glutathione oxidized per milligram of protein per minute [38]. The effectiveness of SOD in inhibiting the auto-oxidation of adrenaline to adrenochrome was measured using the epinephrine auto-oxidation method [43]. Using the Sedlak and Lindsay method, the GR activity in wound tissue was determined [44,45]. GST activity was assayed as described by Habig et al. [46]. A molar extinction coefficient of 9.6 × 10^−3^ was used to calculate enzyme activity. The enzyme activity was measured in milligrams of CDNB-GSH conjugate produced per milligram of protein/min [46].

Lipid peroxidation marker in wound tissue was assessed using thiobarbituric acid (TBA), and estimated the levels of malondialdehyde (MDA) in all treated groups [47]. The rate of lipid peroxidation was presented in micromoles of MDA produced per gram of wet tissue weight/hour.

### 2.9. Statistical Analysis

Statistical comparisons were made between controls and silver nanoparticle-treated groups. Unpaired *T*-tests and two-way analysis of variance (ANOVA) were used to analyze the statistical tests. Data were represented as a mean ± SD for six replicates. A *p* value of less than 0.05 (*p* < 0.05) was considered statistically significant.

## 3. Results

### 3.1. Nanoparticle Treatment Promotes Wound-Healing Efficacy

The wound-healing abilities of CAgNPs and AAgNPs were tested using the excision wound model in Balb/C mice. The synthesis and characterization of the nanoparticles from *C. roseus* and *A. indica* leaf extracts were fully explained in our previous study [34]. The TEM images of both AgNPs were provided as Appendix A. Green-synthesized silver nanoparticles treatment enhanced the wound-healing efficacy, as shown in Figure 1a. The wounds exhibited approximately 93 ± 1% constriction after treatment with CAgNPs and 86 ± 1% closure after treatment with AAgNPs on the 11th day. In contrast, the control wound exhibited approximately 75 ± 1%, vehicle control showed 76 ± 1% wound constriction and positive control treatment showed 86 ± 1% wound constriction. The percentage of wound healing in CAgNPs and AAgNPs treated groups was significantly (*p* < 0.05) higher than that of control and vehicle control groups (Figure 1a). Further supporting evidence for the wound-healing efficacy of AgNPs is provided as Appendix A. These results indicate that green-synthesized AgNPs are efficient wound-healing agents.

In mammals, collagen is the key component and provides integrity to the connective tissue. To determine whether the enhanced wound-healing activity of AgNPs is due to the increased levels of collagen, the quantity of collagen in wound tissue was estimated with hydroxyproline. As depicted in Figure 1b, the hydroxyproline content was relatively low in control (6.9 μg) and vehicle control (5.5 μg) skin samples. However, we found significantly (*p* < 0.05) higher hydroxyproline content in the CAgNPs-treated sample (7.7 μg), which is close to betadine treated skin samples (8 μg). Hydroxyproline in the AAgNPs-treated group (7.1 μg) was significantly (*p* < 0.05) higher compared to the vehicle control-treated group (Figure 2b). Furthermore, the total DNA and protein contents were also significantly (*p* < 0.05) increased in CAgNPs- and AAgNPs-treated tissue samples compared with control vehicle control groups. The total DNA content values were 36.2, 35.5, 38.7, 42.6, and 41.4 μg per 100 mg tissue in control, vehicle control, positive control, CAgNPs, and AAgNPs groups, respectively (Figure 1c). The total protein content in control was 0.31 mg, in vehicle control was 0.33 mg, in positive control was 0.42 mg, in CAgNPs was 0.52 mg, and in AAgNPs was 0.51 mg per 100 mg tissue, as depicted in Figure 1d.

### 3.2. Histopathological Evidence

#### 3.2.1. CAgNPs and AAgNPs Promote Restoration of Epithelium and Dermis (H&E Staining)

H&E staining was performed to evaluate the skin tissue morphology using light microscopic examination. We noticed near to normal epithelialization, mononuclear or inflammatory cells infiltration, mild granulation tissue, and adnexa restoration of the dermis of the control group. In the betadine-treated group, a complete restoration of epithelium, moderate tissue granulation, proliferation, and moderate adnexa restoration of the dermis was observed. In the vehicle control group, near to normal epithelialization, inflammatory cells infiltrates, and mild adnexa restoration and granulation tissue were noticed. Noteworthily, CAgNPs (1% *w*/*w*)-treated dermis exhibited complete restoration of the epithelium (squamous cell epithelium) compared with control and vehicle control groups. In addition, moderate to high granulation tissue, mild inflammatory cell infiltration, and adnexa restoration of the dermis was evidenced with CAgNPs. The AAgNPs (1% *w*/*w*) treated group also showed effective restoration of the epithelium (squamous cell epithelium) compared with control and vehicle control groups. Similarly, moderate to high granulation tissue, mild inflammatory cell infiltration, and adnexa restoration of the dermis were seen in AAgNPs-treated skin samples (Figure 2). The high percentage of wound closure with AgNPs treatment that was gradually increased by treatment duration further explained the effective wound-healing efficacy of AgNPs, and this appears to be better than positive control treatment (Appendix A).

#### 3.2.2. CAgNPs and AAgNPs Enhance Collagen Deposition in Healing Wounds

Collagen deposition is one of the prerequisites for the closure of wounds. Masson’s trichrome stain was used for muscles, cytoplasm, and intercellular fibers. Keratin in red and collagen and nuclei in blue were observed [48]. Animals topically treated with CAgNPs and AAgNPs represented significant collagen deposition, cell infiltration, epithelialization, fibroblast proliferation, and neovascularization, while inflammatory macrophages were fewer in numbers. Fewer macrophages imply an improved inflammatory system during the healing process [49]. Both CAgNPs and AAgNPs showed coarse and fine collagen deposition in wound skin with the moderate proliferation of squamous cell epithelium, fibrous tissue, and adnexa restoration (Figure 3). Images from positive control showed mild collagen deposition and vehicle control showed less collagen deposition in the wound area on day 11 (Figure 3).

### 3.3. CAgNPs and AAgNPs Exhibit Potent DPPH Scavenging Activity

As shown in Figure 4, both AgNPs exhibited potent DPPH radical scavenging ability in a concentration-dependent manner. Quercetin, a known antioxidant, was used as a positive control in DPPH radical scavenging assay [50]. This free radical scavenging activity of CAgNPs and AAgNPs may be due to their strong hydrogen-donating property (reduction of Ag+ ions). The percent of DPPH scavenging activity was 33, 34, 52, 54, and 61% for CAgNPs, while it was 28, 43, 47, 52, and 70% for AAgNPs with the concentrations of 10, 20, 30, 40, and 50 µg/mL, respectively. In nanoparticle comparison, a high concentration of AAgNPs showed slightly greater DPPH scavenging activity (70%) than that of CAgNPs (61%) at the same concentration. The IC50 values for CAgNPs, AAgNPs, and quercetin were 36, 35, and 26 µg/mL, respectively, implying that the free radical scavenging activity of green-synthesized nanoparticles was comparable to standard antioxidants.

### 3.4. CAgNPs and AAgNPs Display ABTS Radical Scavenging Activity

The ABTS radical scavenging ability of CAgNPs and AAgNPs is represented in Figure 5. Although lower concentrations of both AgNPs (10 and 20 µg/mL) were ineffective for scavenging the ABTS, we found that 30 to 50 µg/mL concentrations effectively scavenge the ATBS radical. This radical scavenging ability was paralleled with increased concentration of CAgNPs and AAgNPs (30, 40, and 50 µg/mL). The ABTS radical scavenging percentages were 3, 5, 23, 26, and 32 with CAgNPs treatment and 9, 13, 17, 23, and 40 with AAgNPs treatment for the concentrations of 10, 20, 30, 40, and 50 µg/mL, respectively. The IC50 values were 45, 43, and 26 µg/mL for CAgNPs, AAgNPs, and quercetin, respectively (Figure 5).

### 3.5. AgNPs Enhance Antioxidant Capacity and Suppress Lipid Peroxidation in Wound Skin

Antioxidant enzymes play an essential role in the wound-healing process. As we reported potent free radical scavenging activity from in vitro studies, we determined the in vivo antioxidant efficacy of AgNPs in wound tissue. All antioxidant enzymes, including SOD, CAT, GPx, GR, and GST, were significantly (*p* < 0.05) higher in CAgNPs and AAgNPs-treated skin compared with control and vehicle control skin. To be specific, SOD activity in CAgNPs and AAgNPs groups was 8.2 and 8.6 U/mg protein, respectively, which was significantly (*p* < 0.05) higher than that in control (6.2 U/mg protein) and vehicle control (5.8 U/mg protein) samples (Figure 6a). The augmented CAT activity with CAgNPs and AAgNPs application was also significant (*p* < 0.05) as we found higher concentrations of H_2_O_2_ metabolized/per mg protein tissue (Figure 6b). Furthermore, the glutathione family enzymes, GPx, GR, and GST were also increased significantly (*p* < 0.05) in CAgNPs- and AAgNPs-treated groups compared with control and vehicle control groups (Figure 6c–e). Excessive ROS-induced oxidative damage to lipids, known as lipid peroxidation, is typically represented by elevated MDA levels. We found lower MDA levels in CAgNPs and AAgNPs treated wound tissues compared with that of control wounds (Figure 6f). These findings imply that green-synthesized AgNPs suppress lipid peroxidation in wounds, possibly through improved antioxidant status.

## 4. Discussion

In this study, we have demonstrated the wound-healing efficiency and antioxidant activity of green-synthesized AgNPs from *C. roseus* and *A. indica* by an excision wound model in BALB/c mice for the first time. The synthesis and characterization of silver nanoparticles from *C. roseus* and *A. indica* leaf extracts has been described in our earlier studies [34]. Here, we found that topical application of CAgNPs and AAgNPs on wounds accelerated the healing process by rapid closure of wound area, increased collagen deposition, and re-epithelialization compared to control. We further noticed a significant improvement in skin antioxidant status represented by increased SOD, CAT, GPx, GR, and GST activities. This in vivo antioxidant property was supported by in vitro free radical scavenging activity (DPPH and ABTS) of CAgNPs and AAgNPs. The wound-healing property of both green-synthesized AgNPs might be associated with their potent antioxidant activity.

Studies on the usage of phytotherapeutic agents in wound healing and identifying the underlying mechanism are continuously underway. Plant extracts are known to inhibit oxidative stress and inflammatory response. A recent review emphasized that free radical scavenging, anti-inflammatory, and antimicrobial properties of natural products contribute for their wound-healing efficacy in various animal models [51]. Owing to its antimicrobial and antiseptic property, silver nitrate has been widely used as a cauterizing agent. When silver nitrate is topically applied on wounds, it delivers free silver ions to the tissue that forms an eschar by binding to tissue and obstructing vessels [52]. A study showed faster healing of wounds with hydrocortisone treatment compared to silver nitrate treatment [53]. Thus, silver nanoparticles combined with phytotherapeutic agents may accelerate the wound-healing process by activating various key molecules involved in the healing process. It has been shown that AgNPs-enhanced keratinocyte proliferation and migration to damaged wound sites are prerequisites for wound healing [33]. Biosynthesized AgNPs from parsley, corn silk, and gum arabic exhibited promising antioxidant, anti-inflammatory, and antimicrobial activities [54], which are crucial in the wound-healing process. Green-synthesized AgNPs from *Annona squamosa* increased fibroblast differentiation to myofibroblasts, thus enhancing wound-healing capacity [55]. Mondal et al. summarized tissue regeneration and the wound constriction of injured sites [56]. Other studies also showed that green-synthesized AgNPs treated groups have well-developed collagen fibers with developed blood vessels [57] and enhanced angiogenesis [58].

In our study, biochemical analyses were performed from the isolated wound tissue, and the changes in total collagen, DNA, and protein contents were compared with control and vehicle control-treated groups. The biomolecules, especially DNA and proteins, play a vital role in regulating the wound-healing mechanism. Functionally, these biomolecules regulate the production and release of individual inflammatory cytokines as well as activating the antioxidant signaling cascades in different steps of the wound-healing process. The entire wound-healing process can be influenced by proteins through their role in DNA synthesis, immune system, collagen formation, epidermal growth, and keratinisation [57]. The supplementation of compound protein to mice effectively accelerates the wound-healing process. This was represented by increased collagen deposition, vascularisation, epithelialization, and decreased pro-inflammatory cytokines [59]. Increased DNA and protein contents upon AgNPs application in our study may support the subsequent increase of collagen deposition and the wound-healing process. Hydroxyproline concentration is frequently used as a positive marker for wound healing [60,61] since hydroxyproline concentration intensification can indicate any alteration of collagen synthesis and reflects the process of wound healing in damaged tissues. Kwan et al. demonstrated that the enhanced tensile strength in the wound-healing process of mice treated with green-synthesized AgNPs is due to organized collagen fiber formation [62]. Enhanced hydroxyproline content is a marker of collagen deposition. Pang et al. demonstrated that hydroxyproline is stable in collagen and indirectly represents collagen content [63].

Hydroxyproline is a major constituent and marker for collagen protein present in the tissues, giving firmness, and proline also plays a significant role in this mechanism of collagen genesis [64]. CAgNPs and AAgNPs enhanced the amount of hydroxyproline during wound healing. Collagen formation plays an important role in tissue integrity and helps to restore anatomic function and structure [65]. An increased collagen content helps to induce the wound tensile, which is the core component for producing connective tissue [66]. The histopathological investigation helps to evaluate the wound-healing process and determine the change in the period of epithelization, wound constriction, infiltrates, blood vessels, collagen formation, and fibroblast aggregation. They increased wound-healing efficacy works with the topical application of AgNPs by increased collagen deposition, epithelisation, neovascularization, and fewer macrophages. The collagen formation observed in the images corroborates the increased hydroxyproline content in the AgNPs treated skin. Macrophages are the key regulators in all phases of the wound-healing process and infiltrate the wound area to clear the debris, bacteria, and dead cells, and to reconstruct the damage. Macrophages can change their phenotype from pro-inflammatory to anti-inflammatory during each stage of the wound-healing process [49]. In our study, fewer macrophages observed in CAgNPs- and AAgNPs-treated dermis on day 11 imply the anti-inflammatory property of green-synthesized nanoparticles. However, further evidence is yet to be established to confirm the anti-inflammatory property of CAgNPs and AAgNPs during the healing process.

To evaluate the radical scavenging ability of CAgNPs and AAgNPs, we used DPPH and ABTS radical scavenging assays. DPPH reaction occurs through donating a hydrogen atom and transferring electrons to antioxidants [38]. Both nanoparticles showed potent DPPH scavenging activity, which is comparable to standard antioxidant quercetin. This radical scavenging activity is due to the transfer of hydrogen from AgNPs to free radicals for producing stable products [67] and/or due to the presence of phytochemicals. The ABTS assay was performed to check the efficiency of antioxidants for scavenging the ABTS and giving the ABTS radical (ABTS•+). The green/blue color of ABTS•+ was formed from the ABTS reaction, which gives radical cations [68]. AAgNPs showed relatively higher ABTS radical scavenging activity than CAgNPs at higher concentrations. The ABTS scavenging activity was only 32% for CAgNPs and 40% for AAgNPs at 50 µg/mL. These findings revealed that both *C. roseus* and *A. indica* nanoparticles possess potent free radical scavenging activity, and further studies can be performed on animal models.

Another important finding from our study is that the antioxidant enzyme activities, including SOD, CAT, GPx, and GST, were substantially increased in the cutaneous wound of CAgNPs- and AAgNPs-treated groups. It is documented that the normal physiology of wound healing depends on low concentrations of intracellular ROS and ROS-mediated oxidative stress [18]. Typically, increased antioxidant status inhibits the production of toxic ROS and subsequently inhibits oxidative damage in the wound areas, thereby promoting healing [16]. It has been stated that uncontrolled oxidative stress impairs the wound-healing signaling cascades by altering the functions of cells and molecules involved in the healing process [32]. Therefore, inhibiting ROS production and ROS-induced oxidative damage to lipids and proteins with antioxidant phytochemicals would be an effective strategy to accelerate wound healing. A partial decrease of MDA levels or inhibition of lipid peroxidation with AAgNPs and CAgNPs treatment may explain the rapid wound-healing process in our study. This was further supported by increased skin antioxidant properties following the topical application of CAgNPs and AAgNPs. Hajji et al. reported that chitosan–silver nanoparticles accelerate the wound-healing process in rats through a significant reduction of MDA content and concomitant increase of antioxidant enzyme activities [61]. Green-synthesized nanoparticles possibly enhance wound healing through several mechanisms, including anti-inflammatory, anticancer, and anti-bacterial activities [30,31]. Medicinal plants with AgNPs, which exhibit these pharmacological effects, also possess antioxidant activity and promote wound healing [32]. For instance, the topical application of *A. indica* on wounds improved wound healing through increased antioxidant (SOD, GSH) and anti-inflammatory properties in rats [69]. The leaf extracts of *A. indica* contain a rich amount of quenchers, redox agents, and hydrogen donors that could trigger the antioxidant enzyme activities or inactivate the free radicals [28]. The phytochemicals present in *A. indica*, and *C. roseus* eventually inhibit free radical production and/or quench the elongation of peroxidation chain reactions. Our findings are consistent with the previous reports that showed that SOD, catalase, GPx, and GST activities were increased, whereas lipid peroxidation levels were decreased in wound tissue on the palm with vitamin E and α-tocopherol treatment [70].

Previous findings from our laboratory have shown the characterization of green-synthesized silver nanoparticles from *C. roseus* and *A. indica* plant leaf extracts. As described, UV–Vis spectroscopy revealed a sharp peak at 315–360 nm for *C. roseus* and 410–440 nm for *A. indica* nanoparticles [34]. The XRD spectrum patterns were considerably associated with (113), (111), (124), and (240). Furthermore, the lattice planes in the XRD spectrum were confirmed and cross-checked with the standard referral peak values (JCPDS Files no. 84-0173 and 04-0783), demonstrating that the synthesized AgNPs were crystalline cubic structures in nature. A comparative study of the FTIR spectrum of the control showed the most critical signal peaks of ~3433 cm^−1^, 1632 cm^−1^, and 1384 cm^−1^. These peaks illustrate the presence of the N–H group on the surface of AgNPs, and these visible signals represent the flavonoids and terpenoids that are specifically present in leaf extracts of *C. roseus* and *A. indica*. The average particle size of CAgNPs was 110 nm, with a range of 80–250 nm, and the average particle size of AAgNPs was 60 nm, with various fields of 40–80 nm estimated by DLS. The TEM images showed that the green-synthesized AgNPs were agglomerated and embedded in a dense, thick pattern, likely acting as stabilizing chemical constituents in the leaf extracts of both *C. roseus* and *A. indica.* As presented in Appendix A, the TEM images of CAgNPs revealed particle size variations ranging from 20 to 50 nm [34]. Collectively, these green-synthesized AgNPs from *C. roseus* and *A. indica* with established structure and characterization may play a vital role in improving the antioxidant status (in vitro and in vivo) that endorse wound-healing properties in mice.

## 5. Conclusions

Our findings demonstrated that green-synthesized AgNPs showed efficient antioxidant properties in wound healing. The in vivo wound-healing experiments in BALB/c mice showed improved wound-healing efficacy following CAgNPs, and AAgNPs treatments and were associated with increased hydroxyproline, DNA, and protein contents. The histological evidence represented a decreased scar width, fewer inflammatory cells, well-developed collagen fibers, capillaries, and fibroblasts. Furthermore, antioxidant enzyme activities were increased, and the lipid peroxidation marker decreased in CAgNPs- and AAgNPs-treated skin samples. Our results conclude that green-synthesized AgNPs from *C. roseus* and *A. indica* leaf extracts are potential antioxidant compounds for treating cutaneous wounds and could be used to develop suitable wound-healing medicine or ointment.

## Figures and Tables

**Figure 1 pharmaceutics-15-01517-f001:**
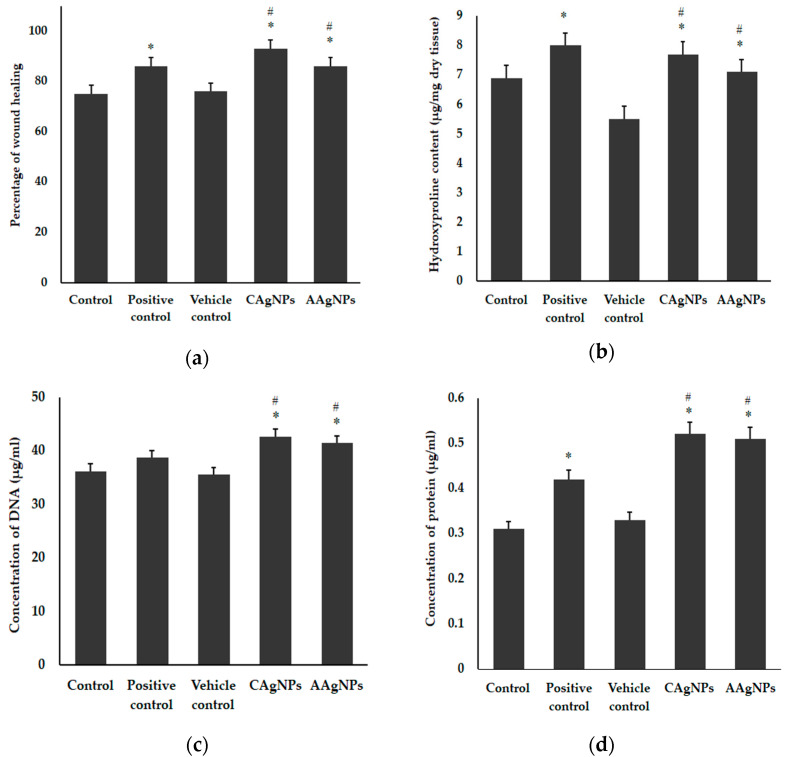
Wound-healing efficacy and biochemical analyses of wound tissues treated with green-synthesized AgNPs. (**a**) Percent of wound-healing efficacy on day 11; (**b**) collagen content (hydroxyproline) from dry wound tissue; (**c**) total DNA content from the wound tissue and (**d**) total protein content in the wound tissue. Values expressed in mean ± SD for six replicates. * indicates significantly different from control (*p* < 0.05), and # indicates significantly different from vehicle control (*p* < 0.05).

**Figure 2 pharmaceutics-15-01517-f002:**
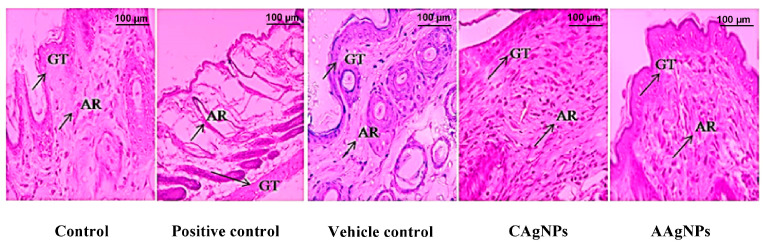
Photomicrograph of cross-section of BALB/c mice wounded skin on day 11 (40× magnification). H&E staining of skin sections of all experimental groups. The arrow indicates granulation tissue (GT) and adnexa restoration (AR) in the respective groups.

**Figure 3 pharmaceutics-15-01517-f003:**
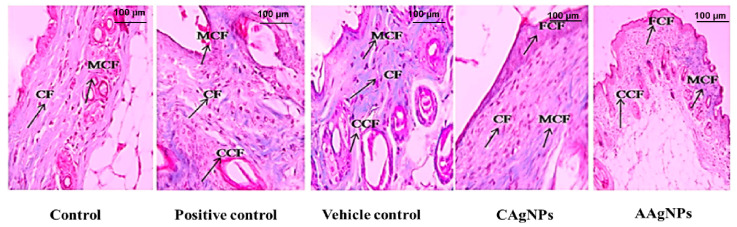
Skin cross-section of wound tissue of BALB/c mice (40× magnification). Masson’s trichrome staining of skin sections of all experimental groups. The arrow indicates collagen fibers (CF), coarse collagen fibers (CCF), mild collagen fibers (MCF), and fine collagen fibers (FCF) in respective groups.

**Figure 4 pharmaceutics-15-01517-f004:**
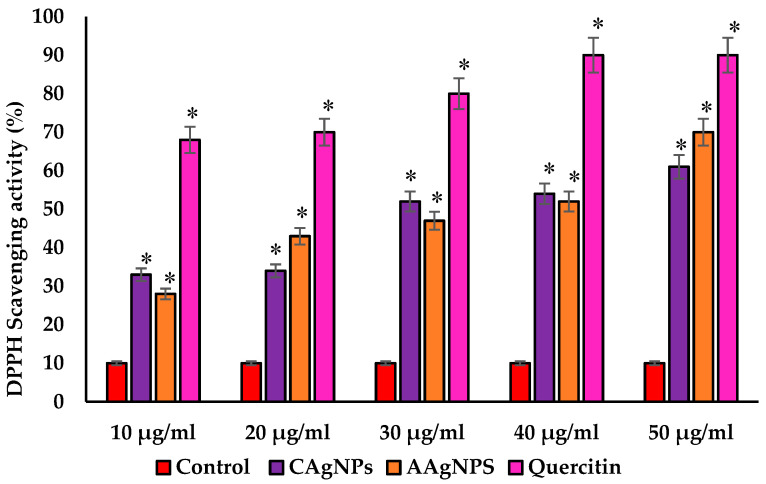
The DPPH radical scavenging activity was shown as a percentage with different concentrations of CAgNPs, AAgNPs, and quercetin. Values expressed in mean ± standard errors. * Significant results were compared with control at *p* < 0.05.

**Figure 5 pharmaceutics-15-01517-f005:**
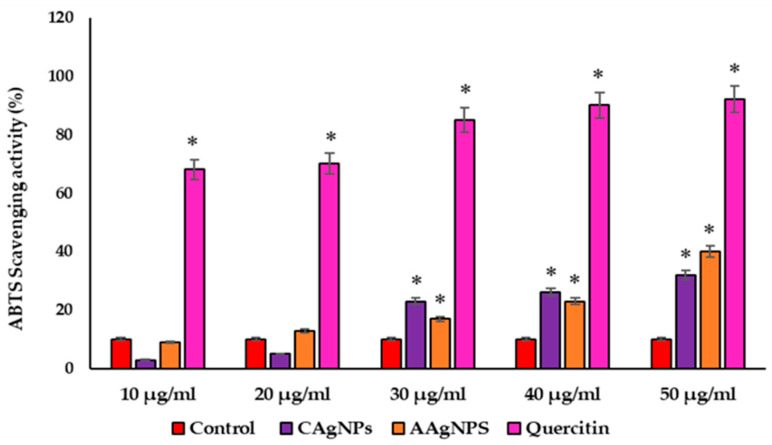
The ABTS radical scavenging activity was showed as a percentage with different concentrations of CAgNPs, AAgNPs, and quercetin. Values are expressed in mean ± standard error. * Significant results were compared with control at *p* < 0.05.

**Figure 6 pharmaceutics-15-01517-f006:**
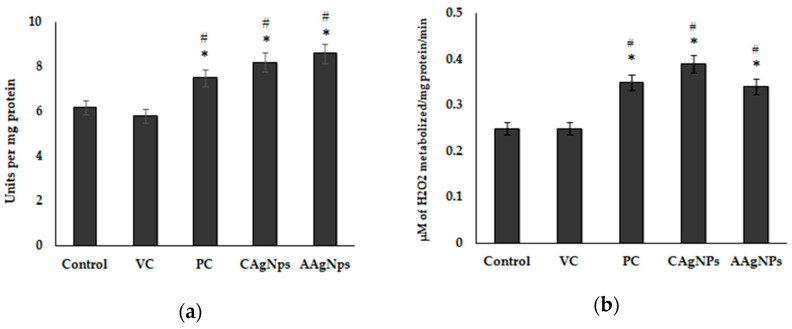
Antioxidant activity of CAgNPs and AAgNPs wound tissue of control, VC (vehicle control, vaseline), PC (positive control, betadine), CAgNPs (*C. roseus*), and AAgNPs (*A. indica*)-treated groups. (**a**) Superoxide dismutase ac, (**b**) catalase, (**c**) glutathione peroxidase, (**d**) glutathione reductase, (**e**) glutathione-s-transferase activities, and (**f**) lipid peroxidation levels. Data are represented as mean ± SD of six independent experiments. * indicates significantly different from control (*p* < 0.05), and # indicates significantly different from vehicle control (*p* < 0.05).

## Data Availability

All the original data of this study are available with the authors.

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
