# Peer review of "Antioxidant Efficacy of Green-Synthesized Silver Nanoparticles Promotes Wound Healing in Mice"

_pharmaceutics, 2023, doi:10.3390/pharmaceutics15051517_

Round 1
Reviewer 1 Report
In the manuscript “Antioxidant efficacy of green synthesized silver nanoparticles from promotes wound healing in mice” Lakkin et al. evaluated antioxidant activities of silver nanoparticles from Azadirachta indica (AAgNPs) and Catharanthus roseus (CAgNPs) leaf extracts in vivo. The research was conducted thoroughly and the manuscript is overall well-written and organized. However, some essential questions regarding the controls, statistical robustness, and results interpretation need to be addressed to support the main article's conclusions.
The following my concerns with the manuscript.
Major concerns:
1- In the 3.1 results section (line 200-219) the author discusses the synthesis of nanoparticles and their characterization. Its highly desirable to present new data for the detailly discussed results (possible as supplementary) or move the summary of the previously published article to the discussion section giving the readers clear view of the new data presented in the article.
2- Supplement representative images of wound healing will be a good addition to support data in Figure 2a.
3- At figure 2 the authors performed important and thorough in vivo experiment testing wound tissues treated with green synthesized AgNPs. Its important to perform statistical analysis of the presented data especially the difference synthesized particles vs vehicle control to support the claim at line 232 and article title.
4- The paper presents good histopathological data to support in vivo results. To highlight the found effect please perform calculated healing score and summarize the parameters that improved when compared to vehicle controls as well as number total samples analysed.
5- Additional parameters such as vascularization and scab formation can be analysed from the slides to support article conclusions,
6- Please add quantification to support line 293 – “while inflammatory macrophages were fewer in number” as well as addressing the inflammation aspect of wound healing.
7- The paper will also benefit from image analysis for quantification of collagen deposition presented in Figure 4.
8- In the first in vivo experiments betadine was used as positive control, please explain this selection and change to Quercetin at figure 5.
9- The author presented no silver nanoparticles control in the experiments, please add data that demonstrated that the effect is not resulting from silver nitrate effect on the tissue.
10- The level of lipid peroxidation presented for synthesized nanoparticles in figure 7f seems to be similar effect of Vaseline - VC. Please detailly address this in results analysis and conclusion made from the data. This is not supporting phrases such as line 380 “We further noticed a substantial increase of skin antioxidant status, and s significant decrease of lipid peroxidation in
CAgNPs and AAgNPs treated mice”, discussion line 378 and conclusion line 456.
Minor concerns:
1- Please rephrase line 76-75 to explain what are “synthesized from physical and chemical s methods”
2- Line 138 – remove e in “ketamin e(50 mg/ml)”
3- Edit article for typographical and punctuation errors

Author Response
Responses to the Reviewer’s Comments – Pharmaceutics# 2329947
Reviewer 1
We express our sincere thanks to the Reviewer for taking time to review our manuscript. All the comments and suggestions provided by you are very useful. We believe that the revisions made in our manuscript according to the comments have improved the quality of our manuscript. All corrections done in the revised manuscript were marked in ‘red’ and here we are providing our point-by-point response to each comment.
Major concerns:
Comment 1: In the 3.1 results section (line 200-219) the author discusses the synthesis of nanoparticles and their characterization. Its highly desirable to present new data for the detailly discussed results (possible as supplementary) or move the summary of the previously published article to the discussion section giving the readers clear view of the new data presented in the article.
Response: Authors are thankful to the Reviewer for this thoughtful comment. We do agree that moving of characterization data and description would allow readers to clearly understand the present study findings. Therefore, we have moved the Figure 1 data as Supplementary Figure S2, and the appropriate description of this Figure is now placed in the Discussion section, Page 12 and 13 of the revised manuscript.
Comment 2: Supplement representative images of wound healing will be a good addition to support data in Figure 2a.
Response: We appreciate the Reviewer for this supporting comment. As suggested, further evidence (representative images and histogram) to support the wound healing efficacy of silver nanoparticles were provided as Supplementary Figure S1A and S1B. The appropriate explanation was included in the Results section, Page 5 of the revised manuscript.
Comment 3: At figure 2 the authors performed important and thorough in vivo experiment testing wound tissues treated with green synthesized AgNPs. Its important to perform statistical analysis of the presented data especially the difference synthesized particles vs vehicle control to support the claim at line 232 and article title.
Response: We are highly thankful to the Reviewer for pointing out the statistical comparisons. In fact we have performed the statistical analysis, but we didn’t label the significance properly in Figure 2. Now we have labeled the significance in Figure 2, and the comparisons were clearly explained in Figure 2 legend. The suitable sentence was incorporated in the Methods section, Page 5. In addition, the statistical comparisons between vehicle control and nanoparticles treated groups have been fully explained in the Results section, Page 6 and 7 of the revised manuscript.
Comment 4: The paper presents good histopathological data to support in vivo results. To highlight the found effect please perform calculated healing score and summarize the parameters that improved when compared to vehicle controls as well as number total samples analysed.
Response: We agree with the Reviewer’s opinion that calculating wound healing scores will provide better comparisons between the groups, and can understand which group exhibited effective healing efficacy. In this study, we haven’t calculated the wound healing scores due to some limitation, and we are apologizing for this. Instead of this, we calculated the percentage of wound closure for all groups on respected days following treatment. These results are now provided as Supplementary data Figure 1B, explaining that percentage of wound healing is considerably higher with CAgNPs and AAgNPs treatment. This wound healing efficacy with nanoparticles was even better than positive control samples. Now this statement was included in the revised manuscript, Page 6 and 7.
All the assays were performed for six samples, and results nanoparticles treated groups were compared with control and positive control groups. This information was given in the revised manuscript wherever applicable.
Comment 5: Additional parameters such as vascularization and scab formation can be analysed from the slides to support article conclusions,
Response: Authors are thankful to the Reviewer for this comment, and addressing this will strengthen our manuscript quality. Scab formation was observed during our in vivo wound healing studies. As suggested, now we included the scab data as Supplementary Figure 1A, and results were appropriately explained in the Results section. However, we regret to inform that we are unable to provide the vascularization data for now. We do consider your suggestion, and will perform the assays using anti-VEGF antibody IHC staining in our ongoing experiments. We appreciate your understanding.
Comment 6: Please add quantification to support line 293 – “while inflammatory macrophages were fewer in number” as well as addressing the inflammation aspect of wound healing.
Response: We are highly thankful to the Reviewer for this meaning comment. We actually observed the slides under microscope and presented the obvious changes. We are apologizing that we haven’t quantified the macrophage numbers from the slides. This might be our limitation and the same was addressed in our Discussion section, Page 11. On the other hand, we referred the following study to support our findings that make our explanations are convincing.
Krzyszczyk, P., et al., The role of macrophages in acute and chronic wound healing and interventions to promote pro-wound healing phenotypes. Frontiers in physiology, 2018. 9: p. 419.
Comment 7: The paper will also benefit from image analysis for quantification of collagen deposition presented in Figure 4.
Response: We admire the Reviewer’s critical evaluation, and agree that quantification of collagen deposition would further strengthen the evidence. However, we regret to inform that we don’t have such quantification facility in our laboratory. This may not be a valid reason, but we have quantified the collagen content in the wound tissue through hydroxyproline assay. This collagen content data was presented as Figure 2A. Hydroxyproline is the major amino acid constituent of extracellular matrix and a valid marker for the collagen content. Now the results of hydroxyproline and collagen formation were collectively discussed in the revised manuscript Page 11.
Comment 8: In the first in vivo experiments betadine was used as positive control, please explain this selection and change to Quercetin at figure 5.
Response: We would like to bring to Reviewer’s kind notice that wound healing assay was performed on mice (in vivo) and used betadine as a positive control. Whereas data in Figure 5 and Figure 6 are from the in vitro experiments, demonstrating the in vitro free radical (DPPH and ABTS) scavenging activities of nanoparticles.
For in vitro studies we used quercetin, a well established antioxidant as a control and compared the in vitro antioxidant capacity of nanoparticles. In case of in vivo studies, we used betadine because betadine (povidone-iodine) has been widely used as an effective antiseptic and wound healing substance. The wound healing efficacy of the nanoparticles in our study is comparable to the known wound healing agent that is betadine. To avoid the discrepancy of using two different controls in one in vivo study, it is appropriate to use same control/compound to examine and compare the antioxidant properties of AgNPs. Supporting references for betadine use were provided in the Methods section, Page 4 of the revised manuscript.
Comment 9: The author presented no silver nanoparticles control in the experiments, please add data that demonstrated that the effect is not resulting from silver nitrate effect on the tissue.
Response: We do agree with the Reviewers opinion that silver nanoparticles may affect or influence the pharmacological effects of plant extracts. To ensure that the pharmacological effects of plant extracts are not because of nanoparticles, we performed our preliminary studies using only silver nanoparticles (AgNO3) as a control group. We then extended our studies and evaluated the antibacterial property of plant extracts. We have done this study by including AgNO3 alone and leaf-extract without AgNO3, and found no effect.
These data revealed that the pharmacological effects of green synthesized nanoparticles are actually from the plant extracts and not due to silver nanoparticles.
Comment 10: The level of lipid peroxidation presented for synthesized nanoparticles in figure 7f seems to be similar effect of Vaseline - VC. Please detailly address this in results analysis and conclusion made from the data. This is not supporting phrases such as line 380 “We further noticed a substantial increase of skin antioxidant status, and s significant decrease of lipid peroxidation in CAgNPs and AAgNPs treated mice”, discussion line 378 and conclusion line 456.
Response: We regret for not clearly discussing the results of lipid peroxidation and antioxidant enzyme activities in the original version of the manuscript. Although lipid peroxidation with Vaseline tend to decrease, but the decreased lipid peroxidation with CAgNPs and AAgNPs was pronounced than the Vaseline alone group. Furthermore, silver nanoparticles-induced decrease was statistically significant compared with the Vaseline alone induced decrease.
Now we have labeled the significance in Figure 6F, and Results, Discussion and Conclusions were revised accordingly.
Minor concerns:
Comment 1: Please rephrase line 76-75 to explain what are “synthesized from physical and chemical s methods”
Response: The mentioned sentence has been revised for better understanding in the revised manuscript, Page 2.
Comment 2: Line 138 – remove e in “ketamin e(50 mg/ml)”
Response: As suggested the ‘e’ was removed and the spelling was corrected in Page 3. .
Comment 3: Edit article for typographical and punctuation errors
Response: We are thankful to the Reviewer for this comment. As suggested the whole manuscript has been checked carefully and corrected all typographical and punctuation errors.
Reviewer 2 Report
Dear Author
It gives me pleasure to review the manuscript under title "antioxidant efficacy of green synthesized silver nanoparticles from promotes wound healing in mice" in journal of Pharmaceutics.
Actually the work is extended for previous work published in antibiotics under title "Green synthesis of silver nanoparticles and evaluation of their antibacterial activity against multidrug-resistant bacteria and wound healing efficacy using a murine model. Antibiotics, 2020. 9 (12): p. 902."
The obtained results showed efficient radical scavenging activity demonstrated by DPPH and ABT and potent skin antioxidant enzyme activities increased against references and control.
Manuscript had a good results and important for reader who interesting at pharmaceutical knowledge . We recommend publication with minor revision after correction of the following comments:
1- You imported TEM from antibiotic journal without indicating the permission.
In wound healing there is lack of discussion about the important of DNA and Protein determination at healing process
2- All figures should be cleared and reconstruct. Especially Figure 7. Has a very bad resolution.
3- Anti-oxidants evaluations confirmed the AgNPs anti-oxidant activity. There are 3 years between original paper 34 " Lakkim, V., et al., Green synthesis of silver nanoparticles and evaluation of their antibacterial activity against multidrug-resistant bacteria 533 and wound healing efficacy using a murine model. Antibiotics, 2020. 9(12): p. 902" and second part. I suppose the data of blood samples shouldn't store more than 2 weeks in fridge.
4- However the manuscript is concerned on anti-oxidant, the author discusses the healing process in details while antioxidant mechanisms for AgNPs are very short.
5- AgNPs already has antioxidant activity. Could you screen the effect of presence phytochemical alone in anti-oxidant activity to confirm synergistic effect of combination between the two ingredients?
Author Response
Responses to the Reviewer’s Comments – Pharmaceutics# 2329947
Reviewer 2
We are highly thankful to the Reviewer for taking time to review our manuscript. All the comments and suggestions provided by the Reviewer are very useful. We believe that the revisions made in our manuscript according to the comments have improved the quality of our manuscript. All corrections done in the manuscript were marked in ‘red’ and here we are providing our point-by-point response to each comment.
Comment 1: 1. You imported TEM from antibiotic journal without indicating the permission.
Response: We are thankful to the Reviewer for reminding the permission of using the previously published data. The TEM results were actually from our own study, and we have cited our paper to mention that these images are from our previously published articled.
However, to avoid copyright issues, we have moved Figure 1 to Supplementary data and the same was labeled in our revised manuscript. Moving TEM data certainly not influenced the results of the present study. For further strengthen and supporting the present study findings we cited previously published ‘Antibiotics 2020. 9 (12): p. 902’ paper wherever applicable.
From the MDPI copyright instructions, we understand that we are allowed to use part of previously published information with proper citation (https://www.mdpi.com/authors/rights).
Comment 2: In wound healing there is lack of discussion about the important of DNA and Protein determination at healing process.
Response: Authors appreciate the Reviewer for this useful comment. We agree that our original version of the manuscript does not fully discuss the findings of DNA and protein content. As suggested, now we have improved our discussion and the role of DNA and protein in wound healing process have been explained in the Discussion section, Page 10 and 11 of the revised manuscript.
Comment 2: All figures should be cleared and reconstruct. Especially Figure 7. Has a very bad resolution.
Response: Thanks to the Reviewer for this comment. We also noticed the poor quality of the graphs. This is due to the conversion of Figures from one format to another format. Also the there are several subpanels in Figure 7 which limit us to decrease the size of each subpanel. However, now we have improved the quality of all Figures. If necessary, we will provide the original Graphs and images to the journal during production time.
Comment 3: Anti-oxidants evaluations confirmed the AgNPs anti-oxidant activity. There are 3 years between original paper 34 " Lakkim, V., et al., Green synthesis of silver nanoparticles and evaluation of their antibacterial activity against multidrug-resistant bacteria 533 and wound healing efficacy using a murine model. Antibiotics, 2020. 9(12): p. 902" and second part. I suppose the data of blood samples shouldn't store more than 2 weeks in fridge.
Response: We appreciate the Reviewer for his/her careful evaluation of our manuscript. We do agree with the Reviewer’s opinion that storage of blood samples more than 2 week is not advisable. Here we would like to bring to Reviewer’s kind notice that the blood biochemical assays were actually done right after the completion of the study. However, due to COVID-19-related drawbacks, graduation existed student and other tasks, we couldn’t finalize part of the data by that year.
Comment 4: However the manuscript is concerned on anti-oxidant, the author discusses the healing process in details while antioxidant mechanisms for AgNPs are very short.
Response: We express our sincere thanks to the Reviewer for bringing out the important point in our manuscript. We also noticed that we haven’t fully discussed the antioxidant part in our old versions. According to the suggestion, now the entire discussion part was amended and antioxidant mechanism with silver nanoparticles were fully explained. The inputs and revisions can be seen in the Discussion part of the revised manuscript, Page 9, 11 and 12.
Comment 5: AgNPs already has antioxidant activity. Could you screen the effect of presence phytochemical alone in anti-oxidant activity to confirm synergistic effect of combination between the two ingredients?
Response: We do agree with the Reviewer’s opinion that AgNPs actually showed some antioxidant activity. To address this issue, in our previous studies, we included AgNO3 alone and plant extracts without adding AgNO3 and found no effect on antibacterial property. We assume that the antioxidant activity of AAgNPs or CAgNPs in our study should be from leaf extracts of the plant. From our FTIR analysis, we also noticed the presence of various functional groups, flavonoids and terpenoids in both AAgNPs and CAgNPs, which may contributed to their antioxidant property. However, we do consider the Reviewer’s comment that individual screening of phytochemicals in each ingredient would further strengthen our findings, and we will do this in our upcoming experiments.
Reviewer 3 Report
Dear Authors,
Please find the attached document for comments and suggestions.
Kind regards,

Author Response
Responses to the Reviewer’s Comments – Pharmaceutics# 2329947
Reviewer 3
We are highly thankful to the Reviewer for taking time to review our manuscript. All the comments and suggestions provided by the Reviewer are very useful. We believe that the revisions made in our manuscript according to the comments have improved the quality of our manuscript. All corrections done in the manuscript were marked in ‘red’ and here we are providing our point-by-point response to each comment.
Comment: In the title of the paper, the word ‘’from’’ should be replaced by ‘’to’’.
Response: We are apologizing for the typo in the Title. Now we have corrected the title.
In the introduction section:
Comment 1: In line 38, the spelling of nanotechnology is incorrect.
Response: Now the ‘nanotechnology’ was corrected accordingly, in Page 1.
Comment 2: The phrase ‘’wound healing’’ should be replaced with ‘’wounds’’ in line 39.
Response: We are thankful to the Reviewer for this comment. As suggested the ‘wound healing’ was changed to ‘wounds’, in Page 1 of the revised manuscript.
Comment 3: In line 44, the phrase ‘’like actual stages’’ can be rephrased to ‘’through the normal stages’’.
Response: We appreciate the Reviewer for this useful comment. As suggested, now the phrase has been changed to ‘through the normal stages’ in the revised manuscript, Page 2.
Comment 4: In line 73, the spelling of titanium is wrong.
Response: We are apologizing for the typo again. Now the spelling of titanium has been corrected.
Comment 5: In line 77, ‘’physical and chemicals methods’’ should be corrected to ‘’ physical and chemical methods’’.
Response: Authors are thankful to the Reviewer for careful revision of our manuscript. As suggested we have done the correction in our revised manuscript, Page 2.
Comment 6: In 91, the word ‘’migrating’’ must change to ‘’migration’’.
Response: As suggested the word ‘migrating’ has been changed to ‘migration’
In the Materials and Methods section:
Comment 1: The formulae in this section must be written in mathematical form.
Response: We are thankful to the Reviewer for this comment and suggestion. As directed, we have presented both DPPH and ABTS formulae in mathematical format in our revised manuscript, Page 3.
Comment 2: In line 180, and line 210 of results section, cm-1 should be corrected to cm-1.
Response: As suggested the unit ‘cm-1’ has been changed to ‘cm-1’ wherever applicable in the whole manuscript.
In the Results section:
Comment 1: From line 200 onwards, the plant names must be continuously written in italics.
Response: We are apologizing for not checking this in our original version of the manuscript. As suggested, plant names have been changed to italics wherever applicable in the revised manuscript.
Comment 2: In line 208, authors should confirm whether they were explaining XRD or SEM data.
Response: We regret for the confusion. We confirmed cubic crystalline structure of green synthesized silver nanoparticles through XRD. We then performed SEM to support the XRD results, which strongly correlate with XRD data.
Round 2
Reviewer 1 Report
During the revision of the manuscript “Antioxidant efficacy of green synthesized silver nanoparticles from promotes wound healing in mice” Lakkin et al. significantly improved the analysis and representation of the results. After addressing the points below the article can be accepted for publication.
Corrections required:
1- In the 3.1 results section please add a sentence with previous paper reference to introduce the particles synthesized – for people that skip the introduction to understand he context.
2- The wound healing images are an important addition to the paper Please add a scale bar.
3- Supplementary Figure S2 labeled as S1. Please update.
4- Line 248 change “L” to “l”.
5- Please address the possible contribution of AgNO3 alone to wound healing process vs plant extracts in the discussion chapter.
6- Please correct the graphs presented in the paper to provide a uniform appearance of the font type and size at X and Y axis, especially at Figure 6.
7- Thank you for adding statistical data to VC in figure 6f. Based on the presented analysis for lipid peroxidation the particles present no significant difference from vehicle control. This does not support the conclusions made in the article lines 335, 440, 448, 27. Unless additional experimental data regarding lipid peroxidation is provided effect similar to VC can not be considered as effective suppression of lipid peroxidation in wounds.
Please revise the article accordingly.
Author Response
Responses to the Comments – Pharmaceutics# 2329947_R2
Reviewer 1
We are highly thankful to the Reviewer for extending his/her time to review our revised manuscript. All comments provided in the 2nd evaluate are valuable and we agree to address them. These comments further helped to strengthen our manuscript quality. As commented, we have revised our manuscript accordingly, and all the corrections in the manuscript were marked in red. Here we are providing our point-by-point responses to each comment.
Comment 1: In the 3.1 results section please add a sentence with previous paper reference to introduce the particles synthesized – for people that skip the introduction to understand he context.
Response: As suggested, a sentence about nanoparticles synthesis and characterization were included. We have also mentioned supplementary data details as it is appropriate in the following sentence. The changes can be seen the revised manuscript, Page 5.
Comment 2: The wound healing images are an important addition to the paper Please add a scale bar.
Response: We appreciate the Reviewer for this comment and agree that the scale bar is important for the wound healing images. As recommended, now we have added the scale bar for both Figure 2 and Figure 3.
Comment 3: Supplementary Figure S2 labeled as S1. Please update.
Response: Apologizing for the typo. Now we have updated the numbers for all Supplementary figures.
Comment 4: Line 248 change “L” to “l”.
Response: As suggested, the ‘L’ has been changed to ‘l’, in the revised manuscript, Page 7.
Comment 5: Please address the possible contribution of AgNO3 alone to wound healing process vs plant extracts in the discussion chapter.
Response: Authors are thankful to the Reviewer for this useful comment. Now we have discussed the role of nanoparticles alone in wound healing process and compared with plant extracts treatment in the Discussion part, Page 11.
Comment 6: Please correct the graphs presented in the paper to provide a uniform appearance of the font type and size at X and Y axis, especially at Figure 6.
Response: Authors sincerely appreciate Reviewer’s comment on quality of figures. Now we have revised all graphs in Figure 6, where X and Y axis were uniformly labeled.
Comment 7: Thank you for adding statistical data to VC in figure 6f. Based on the presented analysis for lipid peroxidation the particles present no significant difference from vehicle control. This does not support the conclusions made in the article lines 335, 440, 448, 27. Unless additional experimental data regarding lipid peroxidation is provided effect similar to VC can not be considered as effective suppression of lipid peroxidation in wounds.
Response: In fact we sincerely appreciate your careful evaluation, and we do agree with the comment. As the changes in lipid peroxidation were not significantly different between nanoparticles and vehicle control treatments, however the decrease of significant compared to control. Now we have revised this information in the whole manuscript, including in Abstract, Discussion and Conclusions. The revised information is appropriate to discuss the antioxidant activity of nanoparticles.
We do believe that the revised version of the manuscript is suitable for your final decision.